# Reproducibility Study of GNNBoundary: Towards Explaining Graph Neural Networks through the Lens of Decision Boundaries

## Abstract

This study reproduces and extends GNNBoundary, a method for explaining Graph Neural Networks (GNNs) by analyzing decision boundaries between graph classes. GNNBoundary identifies adjacent class pairs and generates boundary graphs to provide insights into model behavior. We evaluate the reproducibility of key claims from the original work, including the identification of adjacent classes, the generation of accurate boundary graphs, and the effectiveness of an adaptive loss function in achieving faster convergence. Besides partly generating successful boundary graphs, our reproduction mostly highlights challenges with training variability and convergence, particularly with the Enzymes dataset. This suggests that GNNBoundary's performance is sensitive to hyperparameter settings and random initialization. In addition, we extend GNNBoundary to handle three-class decision boundaries. While it demonstrated its feasibility, it also highlighted limitations in achieving balanced class separability and convergence. By assessing the abilities of GNNBoundary and the extension, this study contributes to improving the transparency and interpretability of GNN decision boundaries. Our findings emphasize the need for refined loss functions, additional baseline comparisons, and methodological extensions to more complex datasets for improved reliability.

## 1 Introduction

Graph Neural Networks (GNNs) are used in many scientific applications, including computer vision, natural language processing, and chemistry (Wu et al., 2020). GNNs are neural architectures designed to model relationships within graph-structured data (Zhou et al., 2020). Besides their efficiency, the complexity of these GNNs presents significant challenges for their application to real-world problems, highlighting the need for effective explainability methods (Saha & Bandyopadhyay, 2024; Yuan et al., 2022). Such explainability methods often generate feature descriptors, such as explanation graphs, which highlight the most discriminative features for each class on model-level (Chen et al., 2024; Wang & Shen, 2022; Yuan et al., 2020). These approaches prioritize graph-based explanations over symbolic representations and mainly focus on individual predictions or class-level patterns. However, a crucial gap persists in understanding how decision boundaries, which separate class predictions, influence a GNN's behavior (McCradden & Stedman, 2024; Saha & Bandyopadhyay, 2024).

GNNBoundary, proposed by Wang & Shen (2024), introduces a novel approach to address this gap by explaining the decision boundaries between similar graph classes. Their method identifies adjacent classes and generates boundary graphs to represent these decision boundaries by incorporating a novel adaptive boundary loss and dynamic regularization scheduler to improve its effectiveness. These boundary graphs provide valuable insights into how models differentiate closely between related classes, thereby clarifying the reasoning of GNNs and complementing existing explanation techniques.

This study focuses on the reproduction of GNNBoundary, specifically the identification of adjacent class pairs and the generation of accurate boundary graphs. In addition, this research introduces a three-way boundary

graph extension. GNNBoundary provides insights into pairwise decision boundaries but real-world graph data often exhibits more complex characteristics between multiple classes (Li et al., 2018). We hypothesize that three-class boundary graphs can reveal insights about ambiguities or overlaps in class separability that are not apparent in pairwise analyses. With this extension of GNNBoundary, we aim to enhance the performance and reliability of the method towards explaining decision boundaries in GNNs.

## 2 Related Work

Explainability methods in GNNs can be categorized into instance-level and model-level. Instance-level methods provide input-dependent explanations but often lack generalizability and fail to capture global decision-making. Model-level methods, though less explored, aim to elucidate a model's overall behavior by generating graphs that trigger specific predictions (Saha & Bandyopadhyay, 2024; Wang & Shen, 2024).

Notable model-level studies include XGNN (Yuan et al., 2020), which uses reinforcement learning to generate class-specific explanation graphs, and D4Explainer (Chen et al., 2024), which employs diffusion-based models but at high computational costs. GNNInterpreter (Wang & Shen, 2022) avoids auxiliary black-box models by leveraging training data embeddings. However, Vasilcoiu et al. (2024) highlights GNNInterpreter's reliance on domain-specific knowledge, sensitivity to hyperparameters and seeds, and training instability.

Unlike earlier approaches, GNNBoundary prioritizes boundary-focused analysis. Following GNNBoundary, Graphon-Explainer (Saha & Bandyopadhyay, 2024) questions its inability to produce class-specific explanations. Therefore, they proposes a technique, that generates both class-specific explanations and boundary graphs.

## 3 Scope of Reproducibility

The core contribution of this study is to reproduces GNNBoundary (Wang & Shen, 2024) and investigate the claims made. Unlike previous techniques, which focus on class-specific explanations, GNNBoundary proposes a novel technique that addresses the decision boundaries between closely related classes. The authors make the following three claims:

1. **Successfully identify adjacent class pairs:** GNNBoundary introduces an algorithm that correctly identifies adjacent class pairs by estimating the likelihood of a smooth boundary between them in the GNN's embedding space.
2. **Superior to baseline methods:** GNNBoundary is claimed that Quantitative comparisons demonstrates that GNNBoundary consistently outperforms baseline methods in generating boundary graphs, producing results that more closely align with the optimal probability distribution.
3. **Effective near-boundary graph generation:** The model is claimed to effectively generate near-boundary graphs, achieving faster convergence and reducing the risk of getting trapped in local minima. Allowing for more accurate boundary representations, improving generalization, and greater stability in identifying class distinctions.

## 4 Background

A graph $G = (\mathcal{V}, \mathcal{E})$ consists of nodes $\mathcal{V}$ with $N$ nodes and edges $\mathcal{E} \subseteq \mathcal{V} \times \mathcal{V}$, with relationships captured in an adjacency matrix $A \in \{0, 1\}^{N \times N}$. Node features can be represented using a feature matrix $Z \in \mathbb{R}^{N \times d}$, where each row corresponds to a node and $d$ is the number of features per node (Zhou et al., 2020). GNNs refine node representations through message passing and aggregation. This enables the property that a GNN can be used as a graph classifier $f$ with $L$ layers.

**Decision Regions and Boundaries** For a graph classifier, the input graph space and intermediate embedding spaces are partitioned into $C$ decision regions, where each region corresponds to a specific class assigned by the classifier. At each layer $l$, the decision regions are denoted as $\{R_c^{(l)} \mid c \in [1, C]\}$. For any

graph $G \in R_c^{(l)}$, the classifier assigns the class $c = \arg\max_k f_k(G)$, where $f_k(G)$ represents the score of the $k$-th class ($1 \leq k \leq C$) (Karimi et al., 2019). Decision boundaries $\mathcal{B}_{c_1 \| c_2}^{(l)}$ separate these regions in the feature space of graph embeddings. These boundaries are defined as the set of points in the embedding space $H^{(l)}$ where the classifier assigns equal confidence to two classes, $c_1$ and $c_2$, where $p(c) = \sigma(\eta_l(H^{(l)}))_c$ is the softmax probability for class $c$. Here, $\eta_l(H^{(l)})$ maps the embeddings $H^{(l)}$ at layer $l$ into logits, and $\sigma(\cdot)$ is the softmax function. Additionally, the probabilities for classes $c_1$ and $c_2$ are strictly greater than those of any other class ($\forall c' \neq c_1, c_2$) (Karimi et al., 2019; Wang & Shen, 2024). The decision boundary can be formally defined as $\mathcal{B}_{c_1 \| c_2}^{(l)} = \{H^{(l)} \mid p(c_1) = p(c_2) > p(c'), \forall c' \neq c_1, c_2\}$.

**Boundary Graph** A boundary graph $G_{c_1 \| c_2} \in \mathcal{B}_{c_1 \| c_2}$ is a specific graph whose embedding $H^{(l)}(G_{c_1 \| c_2})$ lies near or on the decision boundary $B_{c_1 \| c_2}^{(l)}$. For a pair of adjacent classes $c_1$ and $c_2$, the boundary graph represents a challenging case where the GNN struggles to differentiate between the two classes (Wang & Shen, 2024). These boundary graphs are critical for understanding the decision-making process of GNNs.

# 5 Methodology

GNNBoundary has an open-source implementation available.[1] We use their code framework to reproduce the original results and extend it for our additional experiments.[2] In the following section, we describe the methodology we adopted from Wang & Shen (2024).

## 5.1 GNNBoundary

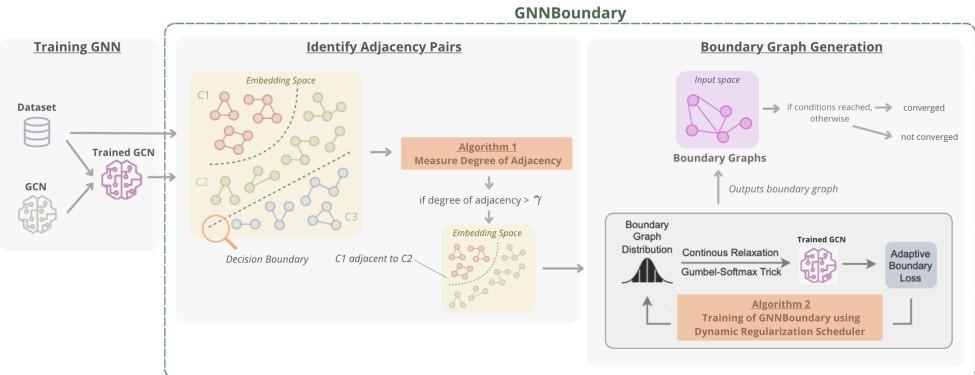

Figure 1: Overview of GNNBoundary from Wang & Shen (2024). A GCN, is trained and then used as input for GNNBoundary to identify adjacent class pairs. These pairs are used to generate boundary graphs for each class pair.

GNNBoundary aims to explain GNNs through the lens of decision boundaries by generating boundary graphs. An overview is presented in Figure 1. Before the GNNBoundary framework, on each dataset a Graph Convolutional Network (GCN) Classifier is trained, which is then used as input for GNNBoundary to identify adjacent class pairs. Utilizing an algorithm designed by the original authors, the degree of adjacency is computed for each class pair. Subsequently, a novel adaptive loss function is introduced that can effectively generate faithful near-boundary graphs for adjacent class pairs. Additionally, GNNBoundary's training ensures adjacent predictions fall within specific probability ranges by using regularization and budget constraints. A dynamic scheduler adjusts these penalties to generate efficient, interpretable boundary graphs.

---

[1] https://github.com/yolandalalala/GNNBoundary
[2] https://anonymous.4open.science/r/FACT_AI_2025-B38D/

### 5.1.1 Identifying Adjacent Classes

A boundary graph represents the decision boundary between two classes $c_1$ and $c_2$. For such a graph to exist, boundary embeddings $\mathbf{H}_{c_1||c_2}^{(l)} \in \mathcal{B}_{c_1||c_2}^{(l)}$ must exist between the embedding regions $\mathcal{R}_{c_1}^{(l)}$ and $\mathcal{R}_{c_2}^{(l)}$. As such, it is important to first identify relevant adjacent class pairs, before attempting to identify a boundary graph $G_{c_1||c_2} \in \mathcal{B}_{c_1||c_2}$. The likelihood of $\mathbf{H}_{c_1||c_2}^{(l)}$ existing is measured by how often boundary embeddings appear between pairs of embeddings $\mathbf{H}_{c_1}^{(l)} = \phi_l(G_{c_1}) \in \mathcal{R}_{c_1}^{(l)}$ and $\mathbf{H}_{c_2}^{(l)} = \phi_l(G_{c_2}) \in \mathcal{R}_{c_2}^{(l)}$, where $\phi_l(G)$ is the embedding of graph $G$ in layer $l$.

The last hidden layer's embedding space ($L-1$-th layer) is used as its features form a linear decision boundary. As such, $\mathbf{H}_{c_1||c_2}^{(l)} \in \mathcal{B}_{c_1||c_2}^{(l)}$ can be found by interpolating between embeddings $\mathbf{H}_{c_1}^{(L-1)}$ and $\mathbf{H}_{c_2}^{(L-1)}$. As a linear decision boundary is used, the interpolation can be doen by determining if the straight-line path between $\mathbf{H}_{c_1}^{(L-1)}$ and $\mathbf{H}_{c_2}^{(L-1)}$ crosses any other decision region $\mathcal{R}_{c'}^{(L-1)}$. Intuitively, this can be understood by looking at the simplified embedding space in Figure 1. A straight path from the embedding of the red $\mathbf{H}_{c_1}^{(L-1)}$ to the embedding blue nodes $\mathbf{H}_{c_3}^{(L-1)}$, would cross embeddings of green nodes $\mathbf{H}_{c_2}^{(L-1)}$, resulting in $\mathbf{H}_{c_1||c_3}^{(l)} \notin \mathcal{B}_{c_1||c_3}^{(l)}$. On the other hand, a straight path from $\mathbf{H}_{c_1}^{(L-1)}$ to $\mathbf{H}_{c_2}^{(L-1)}$ doesn't cross anything, resulting in $\mathbf{H}_{c_1||c_2}^{(l)} \in \mathcal{B}_{c_1||c_2}^{(l)}$. Additionally, classes $c_1$ and $c_2$ are considered adjacent if their degree of adjacency, defined as the frequency of observed boundary embeddings $\mathbf{H}_{c_1||c_2}^{(L-1)}$, exceeds a predefined threshold $\gamma$.

Given the pre-trained GCNClassifier, an algorithm proposed by Wang & Shen (2024), shown in Appendix A, identifies class pairs $c_1$ and $c_2$ where boundary graphs $G_{c_1||c_2}$ are most likely to exist using Monte Carlo sampling. In contrast to the original authors, who use used synthetic graphs generated by GNNInterpreter (Wang & Shen, 2022), we sample graphs directly from the datasets so that our results better represent real-world graph examples. The process to determine adjacency follows:

1. **Sampling Graphs:** Randomly sample $K$ pairs of graphs $G_{c_1} \in \mathcal{R}_{c_1}$ and $G_{c_2} \in \mathcal{R}_{c_2}$.
2. **Interpolating Embeddings:** Interpolate linearly between the embeddings $\mathbf{H}_{c_1}^{(L-1)}$ and $\mathbf{H}_{c_2}^{(L-1)}$.
3. **Checking Decision Boundaries:** Determine if the straight-line path between $\mathbf{H}_{c_1}^{(L-1)}$ and $\mathbf{H}_{c_2}^{(L-1)}$ crosses any other decision region $\mathcal{R}_{c'}^{(L-1)}$. If not, the embeddings $\mathbf{H}_{c_1||c_2}^{(L-1)}$ belong to $\mathcal{B}_{c_1||c_2}^{(L-1)}$.
4. **Computing Adjacency Degree:** Calculate the ratio of successful cases (where $\mathbf{H}_{c_1||c_2}^{(L-1)} \in \mathcal{B}_{c_1||c_2}^{(l)}$) exists) to $K$ total samples.

### 5.1.2 Boundary Graphs Generation and Optimization

Upon the identification of an adjacent class pair $c_1$ and $c_2$, the goal is to output an adequate boundary graph. The graph must satisfy a relaxed near-boundary criterion and adhere to a maximum number of edges. With this goal, the GNNBoundary model learns the probabilistic distribution $P(G)$ that best represents such boundary graphs. The training process converges when the expected graph $E[G]$, derived from the current batch of graph samples $G$ from the distribution, satisfies all boundary graph criteria. GNNBoundary is trained with a loss function that combines an objective function with regularization terms:

$$\text{loss} = w_{objective} \overbrace{\frac{1}{K} \sum_{k=1}^{K} L(G_k)}^{\text{Objective Loss:}} + w_{\text{budget}}^{(t)} \cdot \overbrace{R_{\text{budget}}(\Omega)}^{\text{Budget Penalty}} + \overbrace{w_L \cdot R_L^s(\Omega, Z)}^{\text{Gradient Regularization}} \tag{1}$$

The loss is minimized with respect to parameters $\Omega$ and $Z$, which represent the edge distribution $P(a_{ij})$ and node feature distribution $P(z_i)$, respectively. Through backpropagation and optimization, these parameters are updated to refine both graph structure and node features, which improves how well $E[G]$ aligns with the desired boundary graph characteristics. The following sections detail each loss term.

**Objective Loss Function**   The first term in Equation 1, labeled as Objective Loss, guides the learning of the distribution $P(G)$ for adjacent class pairs $c_1$ and $c_2$. Wang & Shen (2024) define two key properties for boundary graph generation:

1. The posterior probabilities $p(c_1) = p(c_2)$ of sampled graphs should approach 0.5, which reflects a balanced boundary. For a boundary graph $b \in \{c_1, c_2\}$, the objective function should encourage $p(b)$ if $p(b) < 0.5$ and discourage it otherwise, while always discouraging posterior probabilities $p(b')$ for graphs $b' \notin \{c_1, c_2\}$.

2. The logits $f(G)_b$ for $b \in \{c_1, c_2\}$ should be maximized while logits $f(G)_{b'}$ for $b' \notin \{c_1, c_2\}$ should be minimized. The function ensures alignment with target class probabilities while preventing focus on others.

Following the work of the original authors, their alternative objective function is introduced to facilitate the generation of near-boundary graphs. This function serves as a replacement for the cross-entropy loss, which was found to be suboptimal for this task. The proposed objective function is designed to achieve faster convergence while mitigating issues related to local minima (Wang & Shen, 2024).

The formulation enforces class balance by encouraging $p(c_1) = p(c_2) = 0.5$ while ensuring that $f(G)c_1$ and $f(G)c_2$ are not minimized throughout training. The objective function is defined as:

$$\min_G \mathcal{L}(G) = \min_G \sum_{b' \notin \{c_1, c_2\}} \beta f(G)_{b'} \cdot p^*(b')^2 - \sum_{b \in \{c_1, c_2\}} \alpha f(G)_b \cdot (1 - p^*(b))^2 \cdot \mathbb{1}_{p^*(b) < \max_{c \in [1, C]} p^*(c)} \quad (2)$$

where $\alpha$ and $\beta$ are constant hyperparameters. The objective function is differentiated at each iteration to refine a probabilistic graph distribution $P(G)$, enabling the model to produce graphs that increasingly satisfy the near-boundary criterion. The GNNBoundary model assumes graphs follow a Gilbert random graph distribution (Gilbert, 1959) $P(G)$ with independent node features:

$$P(G) = \prod_{v_i \in V} P(z_i) \cdot \prod_{(v_i, v_j) \in \epsilon} P(a_{ij}) \quad (3)$$

where $a_{ij} = 1$ if nodes $v_i$ and $v_j$ form an edge and $a_{ij} = 0$ otherwise, so $a_{ij} \sim \text{Bernoulli}(\theta_{ij})$. Furthermore, $z_i$ represents a categorical node feature of node $v_i$. It is assumed that node features $z_i$ are i.i.d. and follow $z_i \sim \text{Categorical}(p_i)$. As each $a_{ij}$ (i.e., each entry in the adjacency graph A) is binary, P(G) is a discrete distribution over all possible graph configurations, with $\|p_i\|_1 = 1$. However, this presents a problem for differentiation: as matrix A is discrete, $\nabla_A P(G)$ does not exist, meaning gradient-based methods cannot be directly applied via the objective function, and the graphs cannot be optimized. This can be solved by continuous relaxation which in done at the start of very optimization. The pipeline of graph distribution initialization to optimization is described below:

1. **Continuous Relaxation:** To approach this problem and to be able to optimize the discrete graph G using gradients, a differentiable way to represent matrix A is needed. As direct Bernoulli sampling is non-differentiable the concept of continuous relaxation was introduced. Instead of treating edges as discrete, they are defined as a continuous random variable $\tilde{a}_{ij} \in [0, 1]$ parametrized by $\Omega$. Similarly, node features are relaxed to continuous values $\tilde{z}_i \in [0, 1]^d$ parameterized by $Z$, such that $\|\tilde{z}_i\|_1 = 1$. Symbolically this can now be seen as $\tilde{a}_{ij} \sim \text{BinaryConcrete}(\omega_{ij}, \tau_a)$ and $\tilde{z}_i \sim \text{Concrete}(\zeta_i, \tau_z)$, where $\omega_{ij} \in \Omega$ and $\zeta \in Z$ are now learnable parameters, and $\tau_a$, $\tau_z$ are temperature hyperparameters set to 0.15 that control how close the samples are to discrete values. Higher values tend to lead to smoother approximations, while lower values tend to make the samples more discrete.

2. **Reparameterization Via Gumbell-Softmax Trick:** Even though $\tilde{a}_{ij}$ and $\tilde{z}_i$ are differentiable, their sampling must allow gradient backpropagation for the model training. So, it is necessary to replace original Bernoulli and Categorical sampling with a continuous, differentiable sampling process that makes backpropagation possible. This can be done by reparametrization through the Gumbel-Softmax (Concrete) distribution. The Gumbel-Softmax trick provides a way to introduce noise in a controlled way, by drawing a random variable $\epsilon$ for a known distribution. In this case $\epsilon \sim Uniform(0, 1)$. Subsequently, the noise is transformed via a temperature-based softmax or sigmoid to produce a continuous sample, which approximates a discrete outcome. Concretely, after applying this trick each Bernoulli edge $\tilde{a}_{ij}$ and Categorical node feature $\tilde{z}_i$ are:

$$\tilde{a}_{ij} = \text{sigmoid}\left(\frac{\omega_{ij} + \log \epsilon - \log(1 - \epsilon)}{\tau_a}\right), \tilde{z}_i = \text{softmax}\left(\frac{\zeta_i - \log(-\log \epsilon)}{\tau_z}\right) \quad (4)$$

By expressing $\tilde{a}_{ij}$ and $\tilde{z}_i$ as functions of $\omega_{ij}$, $\zeta_i$, and a random noise $\epsilon$, sampling randomness in $\epsilon$ can be isolated, which makes it possible to compute $\frac{\partial \omega_{ij}}{\partial \tilde{a}_{ij}}$ and $\frac{\partial \zeta_i}{\partial \tilde{z}_i}$. If the sampling had been done via a binary method this would have been impossible.

3. **Approximating the Objective with Monte Carlo Sampling:** By expressing $\tilde{a}_{ij}$ and $\tilde{z}_i$ as deterministic functions of $\omega_{ij}, \zeta_i$ and random noise $\epsilon$, the sampling process becomes differentiable with respect to $\omega_{ij}$ and $\zeta_i$. Because $\tilde{A}$ and $\tilde{Z}$ are now continuous, they can be directly substituted into the objective $\mathcal{L}(G)$. This allows standard gradient-based methods to be used to compute $\nabla_{\Omega,Z}\mathcal{L}(\tilde{A}, \tilde{Z})$. Monte Carlo is then employed to sample over $\epsilon$ and approximate the expected loss $\mathbb{E}_{G \sim P(G)}[\mathcal{L}(G)]$. Finally, by minimizing this Monte Carlo estimate we obtain the objective loss term of 1:

$$\min_{\mathbf{A},\mathbf{Z}} \mathcal{L}(G) = \min_{\Theta,\mathbf{P}} \mathbb{E}_{G \sim P(G)}[\mathcal{L}(A, Z)] \approx \min_{\Omega,\mathcal{Z}} \mathbb{E}_{\epsilon \sim U(0,1)}[\mathcal{L}(\tilde{A}, \tilde{Z})] \approx \min_{\Omega, Z} \frac{1}{K} \sum_{k=1}^{K} \mathcal{L}(\tilde{A}^{(k)}, \tilde{Z}^{(k)}) \quad (5)$$

By minimizing this objective function, the model iteratively updates the learned distribution through sampled batches of 32 graphs, with the aim of tuning the distribution so the expected graph E[G] satisfies the boundary graph criteria. This optimization process ensures that the generated boundary graphs exhibit balanced class probabilities, with $p(c_1)$ and $p(c_2)$ converging towards 0.5, thereby aligning with the decision boundary. Additionally, the loss function suppresses logits for non-target classes $c' \notin c_1, c_2$, ensuring that the generated graphs remain within the intended class distinction. The probabilistic graph distribution $P(G)$ is refined through backpropagation, allowing the model to iteratively approximate the near-boundary condition while minimizing the likelihood of convergence to local minima.

**Budget Penalty** The second loss term in Equation 1 is a budget penalty constraint $R_{budget}$, added to encourage graph conciseness. The penalty regularizes the graph size through:

$$R_{\text{budget}} = \text{Softplus}\left(\|\text{sigmoid}(\Omega)\|_1 - B_{loss}\right)^2 \quad (6)$$

where, $B_{loss}$ represents the maximum allowable number of edges in the boundary graph before a penalty is imposed. This constraint discourages the formation of excessively large graphs.

To balance regularization with efficient optimization, the training procedure employs a dynamic scheduler that adaptively adjusts the budget penalty weight $w_{\text{budget}}^{(t)}$ during training. Initially set to $w_{\text{budget}}^{(0)} = 1$, this weight is updated as the graph approaches the near-boundary criterion according to the following rule:

$$w_{\text{budget}}^{(t)} = w_{\text{budget}}^{(t-1)} \cdot s_{\text{inc}}^{\mathbb{I}[\Psi(G^{(t)})]} \cdot s_{\text{dec}}^{\mathbb{I}[\neg\Psi(G^{(t)}) \wedge (s_{\text{dec}} \cdot w_{\text{budget}}^{(t-1)} \geq w_{\text{budget}}^{(0)})]} \quad (7)$$

where $s_{inc}$ and $s_{dec}$ are hyperparameters controlling the rate of increase and decrease, respectively. The penalty is initially small, allowing the model to prioritize learning the overall graph structure and class relationships. As the graph nears the desired boundary, the penalty is gradually increased to enforce graph sparsity and maintain the desired number of edges.

**Gradient Regularization** A third loss term is added in Equation 1, incorporating gradient regularization by adding $L_1$ and $L_2$ regularization terms, denoted by $R_L^s(\Omega, Z)$ for $s \in \{1, 2\}$. The weight $w_L$ represents a weight for $L_1$ as $L_2$ and is defined in Appendix C.

**Training Convergence** The GNNBoundary training procedure has two final convergence criteria:

1. $\Psi(\mathbb{E}[G]) = 1$
2. The size of $\mathbb{E}[G] < B_{stopping}$.

For the first criterion, the model generates boundary graphs using a relaxed near-boundary condition. Since achieving exact boundary conditions where $\sigma(f(G))c_1 = \sigma(f(G))c_2 = 0.5$ is impractical, a graph G approximately belongs to boundary set $\mathcal{B}_{c_1\|c_2}$ when $\Psi(G) = p(c_1), p(c_2) \in [p_{\min}, p_{\max}](G)$. This allows adjacent class probabilities to be approximately equal rather than exactly the same. The second criterion is enforced by the budget penalty loss in Equation 1, where the authors penalize graphs exceeding a strict maximum

number of edges $B_{loss}$. The stopping criterion for the final boundary graph for the maximum number of edges, $B_{stopping}$, is less strict. The training pseudocode for GNNBoundary is presented in Appendix B, and the parameter values are shown in Appendix C.

## 5.2 Datasets

As in the original paper, the GNNBoundary method is evaluated using one synthetic dataset and two real-world datasets[3]. Further, we introduce an additional dataset, MSRC9 (Neumann et al., 2016) [4]. The key characteristics of each dataset are presented in Table 1.

| Dataset | Graphs | Classes | Test Acc. | Avg. Nodes | Avg. Edges |
|---|---|---|---|---|---|
| Collab | 5,000 | 3 | 0.7400 | 74.49 | 2457.78 |
| Motif | 11,531 | 4 | 0.9900 | 57.07 | 77.36 |
| Enzymes | 600 | 6 | 0.5200 | 32.63 | 62.14 |
| MSRC9 | 221 | 8 | 0.9638 | 40.58 | 97.94 |

Table 1: Characteristics of graph datasets

**Synthetic Dataset**    The authors have created a synthetic dataset called Motif, where graphs are classified based on the presence of 4 specific motifs: House (H), House-X (HX), Complete-4 (Cp4), and Complete-5 (Cp5). The generation process of the Motif dataset is explained in detail by Wang & Shen (2022).

**Real-World Datasets**    The authors use two real-world datasets, namely Collab and Enzymes:
- Collab is a scientific collaboration dataset where graphs represent the ego network of a researcher (Yanardag & Vishwanathan, 2015). Nodes represent the researchers and its collaborators, and edges indicate a link between them. Each graph falls into one of three classes, i.e. fields a researcher belongs to: High Energy Physics (HE), Condensed Matter Physics (CM), and Astro Physics (Astro).
- Enzymes is a dataset containing 6 different classes of enzymes (Borgwardt et al., 2005). Furthermore, there are 3 types of nodes that all the enzymes are made up of.

**Additional Dataset**    The introduction of the MSRC9 (Neumann et al., 2016) dataset aims to assess the adaptability of GNNBoundary in handling datasets with a greater number of classes and features. MSRC9 consists of graph-based representations of images derived from the Microsoft Research Cambridge (MSRC) image dataset (Winn et al., 2005). This dataset comprises eight classes, enabling a more extensive evaluation of three-class boundary graphs compared to the Enzymes dataset, as a larger number of three-class decision boundaries can be identified. As a pre-trained classifier is unavailable, a GCN classifier is trained on the MSRC9 dataset using the code framework and the hyperparameters specified in Table 4 in Appendix C.

## 5.3 Experimental Setup and Code

### 5.3.1 Reproducing GNNBoundary

The GNNBoundary methodology described in Section 5.1 and the datasets from Section 5.2 allow us to reproduce the analysis of class adjacency pairs and quantitatively evaluate the generated near-boundary graphs.

**Hyperparameters and Loss Terms**    The GNNBoundary model employs multiple hyperparameters, some of which are specified in the original paper, while others are inferred from the provided demo codebase. However, the codebase contains inconsistencies, such as undocumented parameter choices and loss terms that were rarely used. To ensure reproducibility in our experiments, we adopt a consistent set of hyperparameters, prioritizing values reported in the original paper and supplementing them with the most frequently used values from the codebase. An exception to this is the target size, where the largest value is taken, which is done to prioritize convergence over achieving smaller graphs. A detailed overview of the selected parameters, along with explanations of the final values, is provided in Appendix C.

---

[3] https://drive.google.com/file/d/1O3IRF9mhL2KCCU1eVlCEdssaf6y-pq2h/view?usp=sharing
[4] https://www.chrsmrrs.com/graphkerneldatasets/MSRC_9.zip

**Baseline**   Given the lack of existing methods for comparing decision boundaries in GNNs, the same baseline as proposed in the original study is adopted. This baseline constructs boundary graphs by randomly connecting two sampled graphs from distinct classes ($G_1 \in R_{c_1}$ , $G_2 \in R_{c_2}$) with a randomly assigned edge. This approach is based on the assumption that boundary graphs should encapsulate discriminative features from both classes.

**Identification of Adjacent Class Pairs**   Adjacent class pairs are identified using boundary embeddings to measure the degree of adjacency between classes, as described in Section 5.1.1. For each dataset, a pre-trained GCNClassifier from the codebase performs pairwise boundary analysis. In the experimental study, the sampled graph pairs, $G_{1,k} \in R_{c_1}$ and $G_{2,k} \in R_{c_2}$ are from the datasets. The correct adjacent class pairs are determined using a predefined adjacency score threshold of 0.8, as adopted from the original paper.

**Boundary Graphs Generation and Optimization**   Following Section 5.1.2, GNNBoundary models are trained for each adjacent class pair. For each pair, 1000 near-boundary graphs are generated using near-boundary criteria $p_{min} = 0.45$ and $p_{max} = 0.55$. The number of near-boundary graphs was increased from originally 500 to 1000, to better capture the true underlying distribution. Generated graphs are evaluated to obtain prediction probabilities for the targeted classes. If the model training fails to converge, up to three additional training attempts are made, after which the most recent model is evaluated. It is important to note that the reported probability values are averaged across all graphs. Due to variability in the optimization process and the random seed, these averages are often lower than the best boundary graph results. However, the average provides a more consistent and reliable measure of performance, accounting for fluctuations from random initialization, hyperparameter sensitivity, and stochastic optimization. This better reflects the model's typical behavior across multiple runs, reducing the influence of outliers or overfitting. Using the average ensures the reported performance represents the model's general capabilities, which is important for reproducibility and generalization.

### 5.3.2   Three-class GNNBoundary

While GNNBoundary provides insights into the decision boundaries of GNNs, its formulation inherently assumes that decision boundaries are strictly pairwise. However, in real-world datasets, graph embeddings do not always conform to this assumption. Instead, multiple classes can exhibit overlapping characteristics, resulting in regions where data points lie at the intersection of more than two class boundaries. Building on prior work that demonstrated the existence of three-way decision boundaries in deep neural networks (Li et al., 2018), we introduce three-class GNNBoundary, which incorporates a three-way adjacency perspective. This extension does not merely generalize GNNBoundary but rather proposes an novel alternative lens through which to interpret class interactions in graph classification tasks. By identifying and analyzing these three-class decision boundaries, we aim to capture nuanced ambiguities in class separability that are not apparent in pairwise analyses.

Following the original two-way experimental setup from Section 5.3.1, we extend the adjacency detection method to identify adjacent triplets across the four introduced datasets (Section 5.2). These triplets are then used to generate three-class boundary graphs, with class probabilities aimed at an optimal distribution of 0.33 per class.

**Three-class Adjacency**   Understanding the likelihood of three-class boundary graphs requires defining a three-class adjacency value to quantify the interaction and proximity of the boundaries. Barycentric interpolation extends the pairwise adjacency algorithm of GNNBoundary. It transforms the linear interpolation for two classes into a three-class setting by representing a point as a weighted combination of three reference points (Hormann, 2014). Formally, three-class adjacency exists when an interpolated graph embedding lies at the intersection of the decision regions $\mathcal{R}^{(l)}c_1$, $\mathcal{R}^{(l)}c_2$, and $\mathcal{R}^{(l)}c_3$, where boundary embeddings $\mathbf{H}^{(l)}c_1 \parallel c_2 \parallel c_3$ are located. The interpolation process follows:

$$\mathbf{H}_{interp}^{(L-1)} = a\phi_{L-1}(G_{1_k}) + b\phi_{L-1}(G_{2_k}) + c\phi_{L-1}(G_{3_k}) \qquad (8)$$

where $a, b, c$ are barycentric coordinates ensuring $a + b + c = 1$ The three-way adjacency score is computed as the frequency with which interpolated embeddings are classified into all three adjacent classes over $K$

samples. The adjacency threshold for these triplets was set to 0.9 for most datasets but lowered to 0.6 for Enzymes due to its lower classification accuracy. We validate this three-class adjacency method by comparing it against the pairwise adjacency scores of the same triplet. The algorithm in Appendix E utilizes interpolating between embeddings to estimate this likelihood.

**Training Three-class GNNBoundary** For adjacent triplets, the original GNNBoundary framework from Section 5.1.2 is modified to accommodate three-class decision boundaries. The model generates 1000 near-boundary graphs, with the loss function adapted to an initial probability range of $p_{min}$ is reduced by 0.033 after each failed attempt, for a maximum of five iterations. This incremental reduction gradually relaxes the decision boundary constraints, enhancing the model's ability to adapt to the complexity of three-class boundaries. The initial lower bound of $p_{min} = 0.3$ may be overly restrictive; therefore, reducing it incrementally mitigates abrupt changes while maintaining sufficient flexibility for convergence. If convergence is not achieved after all attempts, the most recent non-converged model is retained.

### 5.4 Computational Requirements

Experiments were conducted on a laptop with AMD Ryzen 7 4000 CPU and Nvidia GeForce GTX 1650 TI. For the MSRC9 dataset, GPU and CPU training times were comparable. For the larger Collab dataset, GPU training reduced time from 77.2 to 8.06 seconds per iteration. The complete experiments took 105 minutes, with three-class adjacency extensions requiring 140 minutes. With GPU (20W), CPU (12W), and RAM (2W) power consumption, Netherlands' carbon intensity of 0.564[5], and an assumed PUE of 2, the total carbon footprint was approximately 70.76g $CO_2$, equivalent to driving 0.284 km by car.

## 6 Results

### 6.1 Identifying Adjacent Classes

In our reproduction study, we identified 7 out of 11 adjacent graph classes that were found by the original authors. Figure 2 shows we find similar adjacency values for Collab and Motif, leading to the same adjacent class pairs as the authors. However, for Enzymes, the adjacency values for some class pairs differ from the results. Pairs EC1-EC2, EC2-EC4, and EC3-EC4 show a substantial difference in adjacency compared to the original results. At the chosen threshold of more than 0.8, we found 3, 2, and 2 adjacent class pairs respectively for the Motif, Collab, and Enzymes datasets, while the original paper found 3, 2, and 6. To reproduce the experiment precisely, we create boundary graphs for all the adjacent pairs found by the original authors. For the MSRC9 dataset, 15 adjacent pairs were found. For simplicity, the 6 pairs with the highest adjacency score were used for the GNNBoundary generation.

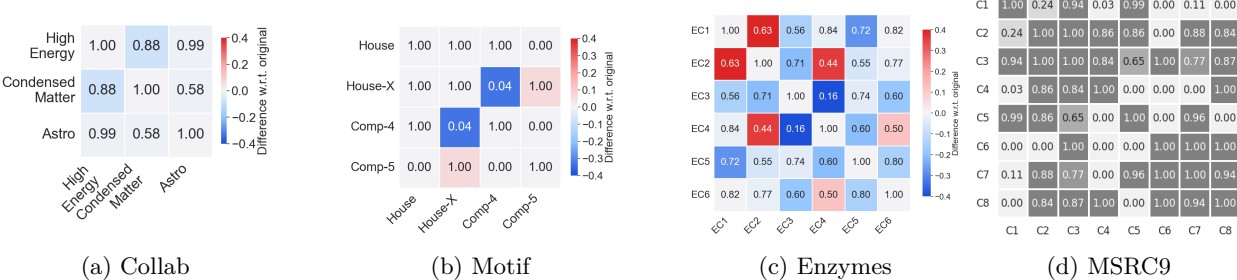

| | | | | |
|---|---|---|---|---|
| (a) Collab | (b) Motif | (c) Enzymes | (d) MSRC9 |

Figure 2: Confusion matrices show reproduced adjacency values between graph classes from four datasets. In *a*, *b*, and *c*, reproduced values that are larger than the than original adjacency scores are more prominently red or blue. For MSRC9 (*d*), as there is no comparison with the authors, the darker shades of gray indicate larger adjacency scores

---

[5]https://www.energyinst.org/statistical-review

## 6.2 GNNBoundary Quantitative Evaluation

To support Claims 2 and 3 in our reproduction, we found that GNNBoundary is able to consistently generate boundary graphs with class probabilities close to the optimal value of 0.5. Table 2 demonstrates that our GNNBoundary reproduction performs well by generating probabilities close to the optimal value of 0.5. However, our results are less consistent, since the original probabilities for the target classes are closer to 0.5 with smaller standard deviations. The differences between our reproduction and the original results are largest in the Enzymes dataset, followed by Collab, with Motif showing the closest alignment to the original findings. Comparison of our results with the baseline method can be found in Appendix D.

| Dataset | $c_1$ | $c_2$ | $p(c_1)$ orig | $p(c_1)$ | $\Delta$ | $p(c_2)$ orig | $p(c_2)$ | $\Delta$ |
|---|---|---|---|---|---|---|---|---|
| Collab | HE | CM | $0.473 \pm 0.015$ | $0.598 \pm 0.148$ | 0.125 | $0.487 \pm 0.016$ | $0.377 \pm 0.153$ | -0.110 |
| | HE | Astro | $0.526 \pm 0.013$ | $0.682 \pm 0.098$ | 0.156 | $0.466 \pm 0.013$ | $0.226 \pm 0.122$ | -0.240 |
| Motif | H | HX | $0.501 \pm 0.028$ | $0.501 \pm 0.165$ | 0.000 | $0.499 \pm 0.028$ | $0.498 \pm 0.167$ | -0.001 |
| | H | Cp4 | $0.498 \pm 0.028$ | $0.476 \pm 0.054$ | -0.022 | $0.501 \pm 0.028$ | $0.529 \pm 0.054$ | 0.028 |
| | HX | Cp5 | $0.491 \pm 0.026$ | $0.242 \pm 0.055$ | -0.249 | $0.509 \pm 0.026$ | $0.325 \pm 0.033$ | -0.184 |
| Enzymes | EC1 | EC4 | $0.489 \pm 0.023$ | $0.188 \pm 0.038$ | -0.301 | $0.487 \pm 0.021$ | $0.222 \pm 0.047$ | -0.265 |
| | EC1 | EC5 | $0.492 \pm 0.023$ | $0.151 \pm 0.030$ | -0.341 | $0.489 \pm 0.025$ | $0.445 \pm 0.044$ | -0.044 |
| | EC1 | EC6 | $0.485 \pm 0.028$ | $0.286 \pm 0.048$ | -0.199 | $0.472 \pm 0.017$ | $0.307 \pm 0.028$ | -0.165 |
| | EC2 | EC3 | $0.488 \pm 0.025$ | $0.434 \pm 0.047$ | -0.054 | $0.488 \pm 0.025$ | $0.523 \pm 0.062$ | 0.035 |
| | EC4 | EC5 | $0.480 \pm 0.024$ | $0.344 \pm 0.107$ | -0.136 | $0.486 \pm 0.024$ | $0.097 \pm 0.080$ | -0.389 |
| | EC5 | EC6 | $0.481 \pm 0.022$ | $0.438 \pm 0.113$ | -0.043 | $0.486 \pm 0.023$ | $0.552 \pm 0.111$ | -0.066 |
| MSRC9 | C5 | C7 | - | $0.497 \pm 0.082$ | - | - | $0.500 \pm 0.083$ | - |
| | C4 | C8 | - | $1.0 \pm 0.0$ | - | - | $3.4e-9 \pm 7.9e-9$ | - |
| | C2 | C3 | - | $0.502 \pm 0.087$ | - | - | $0.473 \pm 0.094$ | - |
| | C7 | C8 | - | $0.470 \pm 0.146$ | - | - | $0.483 \pm 0.127$ | - |
| | C1 | C3 | - | $0.467 \pm 0.089$ | - | - | $0.528 \pm 0.088$ | - |
| | C3 | C6 | - | $0.448 \pm 0.092$ | - | - | $0.474 \pm 0.087$ | - |

Table 2: Comparison of GNNBoundary reproduction results with the original paper's probabilities, showing differences $\Delta$. Each entry reports the mean predicted class probability $\pm \sigma$. Red delta values denote cases where our mean differs from the original by more than $2\sigma$. Probabilities near 0.33 are preferred.

## 6.3 Three-class GNNBoundary

Table 3 presents adjacency and boundary graph generation results for three classes. Appendix F validates three-way adjacency through high pairwise scores. The results show that when GNNBoundary converges, class predictions approach the desired 0.33. However, for some class combinations, non-convergence results in inaccurate boundary graphs.

| Dataset | $c_1$ | $c_2$ | $c_3$ | GNNBoundary | | | Adj. score | Conv. rate |
|---|---|---|---|---|---|---|---|---|
| | | | | $p(c_1)$ | $p(c_2)$ | $p(c_3)$ | | |
| Collab | HE | CM | Astro | $0.405 \pm 0.039$ | $0.379 \pm 0.047$ | $0.216 \pm 0.048$ | 1.0 | 1.0 |
| Motif | H | HX | Cp4 | $0.704 \pm 0.076$ | $0.003 \pm 0.003$ | $0.290 \pm 0.078$ | 1.0 | 0.0 |
| | H | HX | Cp5 | $0.161 \pm 0.004$ | $0.427 \pm 0.031$ | $0.413 \pm 0.026$ | 1.0 | 0.0 |
| Enzymes | EC1 | EC5 | EC6 | $0.030 \pm 0.012$ | $0.322 \pm 0.013$ | $0.399 \pm 0.024$ | 0.72 | 0.0 |
| | EC1 | EC2 | EC6 | $0.152 \pm 0.020$ | $0.215 \pm 0.025$ | $0.156 \pm 0.010$ | 0.71 | 0.6 |
| | EC2 | EC5 | EC6 | $0.289 \pm 0.016$ | $0.222 \pm 0.005$ | $0.192 \pm 0.029$ | 0.69 | 0.0 |
| | EC2 | EC3 | EC6 | $0.338 \pm 0.021$ | $0.355 \pm 0.046$ | $0.240 \pm 0.039$ | 0.63 | 0.8 |
| MSRC9 | C5 | C6 | C7 | $0.367 \pm 0.082$ | $0.333 \pm 0.100$ | $0.299 \pm 0.084$ | 1.0 | 1.0 |
| | C0 | C2 | C4 | $0.198 \pm 0.057$ | $0.614 \pm 0.027$ | $0.184 \pm 0.049$ | 0.98 | 0.4 |
| | C1 | C2 | C3 | $0.307 \pm 0.098$ | $0.314 \pm 0.123$ | $0.374 \pm 0.169$ | 0.97 | 1.0 |
| | C1 | C2 | C6 | $0.225 \pm 0.073$ | $0.206 \pm 0.073$ | $0.298 \pm 0.129$ | 0.94 | 0.8 |
| | C2 | C5 | C7 | $0.337 \pm 0.096$ | $0.295 \pm 0.082$ | $0.265 \pm 0.078$ | 0.94 | 0.8 |
| | C1 | C2 | C7 | $0.440 \pm 0.117$ | $0.159 \pm 0.036$ | $0.064 \pm 0.023$ | 0.92 | 0.2 |
| | C3 | C5 | C7 | $0.325 \pm 0.054$ | $0.006 \pm 0.002$ | $0.669 \pm 0.053$ | 0.92 | 0.0 |

Table 3: Quantitative evaluation of 1000 boundary graphs for three adjacent classes. Reports the mean predicted class probability $\pm \sigma$ over 1000 graphs. Thresholds: MSRC9, Motif, Collab = 0.9; Enzymes = 0.6. Probabilities near 0.33 are ideal.

# 7 Discussion

## 7.1 GNNBoundary Reproduction

This reproduction study evaluated three claims made by the authors of GNNBoundary (Wang & Shen, 2024) as an explanation method for GNN decision boundaries.

For Claim 1, the reproduction of adjacent class identification was partially successful, with 7 out of 11 pairs aligning with the original study. While results for the Collab and Motif datasets were consistent with the original findings, discrepancies were observed in the Enzymes dataset, where only 2 out of 6 adjacent pairs were identified. Initially, we speculated that an increased number of classes could reduce adjacency identification performance; however, the successful identification of adjacent pairs in the MSRC9 dataset contradicts this assumption.

Notably, we observed a correlation between adjacency scores and test classification accuracy: lower accuracy of the GCNClassifier for Enzymes resulted in unclear decision boundaries, whereas higher accuracy for Motif and MSRC9 yielded a clearer distinction between adjacent and non-adjacent pairs. The variability in the adjacency results might, in part, stem from differences in the random sampling process, as the original paper did not specify a fixed random seed. However, given the large number of samples drawn, the effect of the seed alone should be minimal. If randomness indeed played a significant role in the observed discrepancies, increasing the number of sampled pairs even further maybe could have mitigated this effect. Instead, variations are more likely influenced by factors such as hyperparameter sensitivity, training dynamics, or subtle implementation details that were not explicitly documented in the original study. These findings partially validate Claim 1, demonstrating that the proposed algorithm effectively identifies adjacent class pairs while revealing potential sources of instability in the method.

For Claim 2, our reproduction of the quantitative evaluation of GNNBoundary exhibited significant deviations from the original results. For the Collab and Enzymes datasets, 14 out of 16 probability estimates deviated by more than $2\sigma$, whereas in the Motif dataset, only 2 out of 6 exhibited such deviations. Nonetheless, the generated boundary graphs consistently outperformed the baseline methods, as demonstrated in Appendix D. While the baseline, being random, may not be a perfect comparison, it still supports the validation of Claim 2. For MSRC9, 5 out of 6 generated boundary graphs yielded probabilities close to the optimal value of 0.5. We speculate that the clearer identified adjacent pairs in MSRC9 and Motif lead to more accurate boundary graphs.

The deviations in our results can be attributed to the inherent variability in GNNBoundary training. We assume that the original authors reported probabilities from a particularly well-performing model rather than an average over multiple runs, whereas our results are derived from an average of five independent training runs. This methodological difference, along with ambiguities in the codebase and unspecified hyperparameters, contributed to inconsistencies in reproduction. High variance in convergence rates suggests sensitivity to hyperparameters and weight initialization. Additionally, GNNBoundary's complex loss function, with multiple regularization terms and adaptive penalties, likely exacerbates this sensitivity, making convergence more challenging. Small training modifications can lead to significantly different outcomes, with some models failing to escape local minima. The authors suggest an optimal loss function should encourage class logits and balance posterior probabilities around 0.5. While this sometimes holds, our results show inconsistencies, some boundary graphs have imbalanced probabilities, while others retain low posteriors, indicating insufficient logit separation. This suggests the loss function may not be ideally suited the optimization of boundary graph generation. Alternatively, discrepancies may arise from additional loss terms or suboptimal hyperparameter settings. Given the high variance and methodological ambiguities, we cannot fully validate Claim 3, which asserts that the adaptive loss function consistently generates near-boundary graphs with faster convergence and reduced risk of local minima.

## 7.2 Three-class GNNBoundary

The extension of the GNNBoundary methodology to three-way boundary graphs achieved moderate success. The three-way boundary graphs did not yield perfectly balanced splits for all class triplets, and some mod-

els failed to converge altogether. This result is unexpected since the high adjacency score for the triplets should imply that there are graphs on the boundary of the three classes. In addition, our three-way adjacency method is supported by high pairwise adjacency scores within these triplets, shown in Appendix F. However, it could be that our three-way adjacency method does not correspond well enough with the actual decision boundaries between the classes. Another possible cause could be the custom loss function needs further refinement to handle multi-class decision boundaries effectively. Incorporating a more robust multi-class regularization term in the loss function could potentially improve the model's ability to generate accurate three-way boundary graphs. Another option could be that the GNNBoundary training process and its hyperparameters need to be further optimized for the three-way decision boundaries. Although our methodology did not produce perfect three-way boundary graphs, we believe this extension is a step forward towards enhancing transparency and reliability of GNNs in practical applications.

### 7.3   Reflection

The detailed explanations by Wang & Shen (2024) of the mathematical concepts behind the adjacency calculation and the GNNBoundary methodology helped us understand the background and their research well. Additionally, having access to the authors' codebase and datasets provided us with a foundation for reproducing the experiments and further understand the methodology. However, recreating the original experiments was challenging, due to ambiguities in the codebases and its execution. It was unclear whether the authors employed certain functions from the codebase or relied on alternative methods to reduce variability, due to randomness. Additionally, unused code in the demo further complicated understanding of how experiments should be properly executed. The level of uncertainty was increased by the lack of documentation on code structure and hyperparameter choices, leading to numerous discrepancies between the open-source codebase implementation and the paper.

Additionally, we contacted the original authors with questions regarding their methodology, codebase, and hyperparameters, including details on the boundary graph generation process, hyperparameter selection, and undocumented loss terms. However, we did not receive a response.

### 7.4   Future Work

Future work should explore methods to enhance classifier accuracy, particularly for datasets like Enzymes, where lower classification performance may lead to unreliable adjacency estimations. Analyzing the relationship between classifier confidence and adjacency scores could improve the robustness of GNNBoundary by mitigating sensitivity to noisy predictions

Secondly, while our speculation about the influence of class numbers on adjacency identification was disproved by the MSRC9 dataset, it remains a relevant topic. Future studies could explore the impact of increasing class numbers beyond eight, using larger datasets to understand how adjacency scores evolve as the dataset grows. This could help GNNBoundary scale to more complex, real-world applications with many classes.

Introducing additional baseline methods beyond random sampling, such as Graphon-Explainer (Saha & Bandyopadhyay, 2024), could provide more comprehensive comparisons for decision boundary analysis. Additionally, the variability in training outcomes suggests that increasing the sampling size could help stabilize results by capturing a more representative set of boundary graphs. Exploring alternative sampling methods or ensemble techniques would further improve consistency and reliability. Furthermore, systematic exploration of hyperparameter sensitivity through fine-tuning could optimize GNNBoundary's performance, enhancing reproducibility.

The challenges with three-class decision boundaries suggest that future work should focus on refining the loss function to better handle multi-class boundaries. Incorporating robust multi-class regularization or adaptive scheduling could improve convergence and balance in three-class boundary graphs.

Finally, while Wang & Shen (2024) introduced useful metrics like boundary thickness, margin, and complexity, their application was unclear, thereby not explored in this study. Future research could explore and validate these metrics, providing deeper insights into the decision-making process of GNNs and improving the interpretability of decision boundaries.

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

# 8 Appendix

## A Adjacency Algorithm

---

**Algorithm 1** Measure the Degree of Adjacency of a Class Pair

---

1: **count** $\leftarrow 0$
2: **for** $k \leftarrow 1 \ldots K$ **do**
3:      Randomly sample two graphs $G_{1_k} \in \mathcal{R}_{c_1}$ and $G_{2_k} \in \mathcal{R}_{c_2}$
4:      Compute **score** $= \prod_{\lambda \in [0,1]} \mathbb{1}_{\{c_1, c_2\}} \left( \arg\max_c \eta_{L-1} \left( \lambda \phi_{L-1}(G_{1_k}) + (1 - \lambda) \phi_{L-1}(G_{2_k}) \right) \right)$
5:      **if** score $\neq 0$ **then**
6:          **count** $\leftarrow$ **count** $+1$
7:      **end if**
8: **end for**
9: **return** $\frac{\mathbf{count}}{K}$

---

## B GNNBoundary Training Algorithm

---

**Algorithm 2** Training of GNNBoundary using Dynamic Regularization Scheduler

---

Initialize sampler parameters $\Omega$ and $Z$
**for** each iteration $t$ **do**
     Sample a batch of $k$ graphs $\{G_1, \ldots, G_K\}$ from sampler with parameters $\Omega$ and $Z$
     loss $\leftarrow \frac{1}{K} \sum_{k=1}^{K} L(G_k) + w_{\text{budget}}^{(t)} \cdot R_{\text{budget}}(\Omega) + w_L \cdot R_L^s(\Omega, Z)$
     Minimize loss with respect to $\Omega$ and $Z$
     **if** $\Psi(\mathbb{E}[G]) = 1$ and the size of $\mathbb{E}[G] < B_{stopping}$ **then**
         **return** $\Omega$ and $Z$
     **end if**
**end for**

---

## C Hyperparameter Values

| Dataset | Hidden Channels | Number of Layers |
|---------|-----------------|------------------|
| Collab  | 64              | 5                |
| Motif   | 6               | 3                |
| Enzymes | 32              | 3                |
| MSRC9   | 16              | 5                |

Table 4: GCN Classifier architecture parameter values.

| Parameter | Value |
|-----------|-------|
| Max Nodes | 25 |
| Temperature | 0.15 |
| Batch Size | 32 |
| $\alpha$ | 1 |
| $\beta$ | 1 |
| $w_{\text{objective}}$ | 25 |
| $W_{L1}$ | 1 |
| $W_{L2}$ | 1 |
| Budget Penalty: $B_{loss}$ | 10 |
| Target Size $B_{stopping}$ | 60 |
| $s_{\text{inc}}$ | 1.1 |
| $s_{\text{dec}}$ | 0.95 |

Table 5: Unified hyperparameters and loss weight configuration for all class pairs.

In Tables 4 and 5 the parameters are described that we used in our reproduction study, of which the complete procedure can be seen in Figure 1. The values were chosen with the aim of current and future reproduction. Where possible, values were taken from the paper and then supplemented with the most frequently seen

values in the demo codebase. A number of undocumented loss terms were seen in the codebase. However, these were not used in the reproduction, as they were rarely activated, making it unclear when or if they had been used by the authors.

## D Baseline Comparison

| Dataset | $c_1$ | $c_2$ | GNNBoundary | | Baseline | |
|---------|-------|-------|-------------|--------------|----------|--------------|
| | | | $p(c_1)$ | $p(c_2)$ | $p(c_1)$ | $p(c_2)$ |
| Collab | HE | CM | $0.461 \pm 0.050$ | $0.474 \pm 0.050$ | $0.262 \pm 0.169$ | $0.016 \pm 0.071$ |
| | HE | Astro | $0.622 \pm 0.082$ | $0.313 \pm 0.095$ | $0.260 \pm 0.166$ | $0.723 \pm 0.197$ |
| Motif | H | HX | $0.501 \pm 0.165$ | $0.498 \pm 0.167$ | $0.754 \pm 0.149$ | $0.004 \pm 0.018$ |
| | H | Cp4 | $0.476 \pm 0.054$ | $0.523 \pm 0.054$ | $0.756 \pm 0.151$ | $0.240 \pm 0.153$ |
| | HX | Cp5 | $0.242 \pm 0.055$ | $0.325 \pm 0.033$ | $0.003 \pm 0.003$ | $3.00 \pm 4.84$ |
| Enzymes | EC1 | EC4 | $0.188 \pm 0.038$ | $0.222 \pm 0.031$ | $0.031 \pm 0.153$ | $0.007 \pm 0.068$ |
| | EC1 | EC5 | $0.151 \pm 0.030$ | $0.449 \pm 0.044$ | $0.029 \pm 0.143$ | $0.070 \pm 0.242$ |
| | EC1 | EC6 | $0.286 \pm 0.048$ | $0.306 \pm 0.028$ | $0.028 \pm 0.141$ | $0.040 \pm 0.174$ |
| | EC2 | EC3 | $0.434 \pm 0.047$ | $0.523 \pm 0.062$ | $0.091 \pm 0.257$ | $0.745 \pm 0.407$ |
| | EC4 | EC5 | $0.344 \pm 0.107$ | $0.097 \pm 0.080$ | $0.014 \pm 0.101$ | $0.072 \pm 0.245$ |
| | EC5 | EC6 | $0.438 \pm 0.113$ | $0.552 \pm 0.111$ | $0.081 \pm 0.258$ | $0.040 \pm 0.172$ |
| MSRC9 | C5 | C7 | $0.497 \pm 0.082$ | $0.500 \pm 0.083$ | $5.9\text{e-}5 \pm 0.001$ | $0.165 \pm 0.349$ |
| | C4 | C8 | $1.0 \pm 0.0$ | $3.4\text{e-}9 \pm 7.9\text{e-}9$ | $0.056 \pm 0.206$ | $0.645 \pm 0.434$ |
| | C2 | C3 | $0.502 \pm 0.087$ | $0.473 \pm 0.094$ | $0.014 \pm 0.094$ | $0.037 \pm 0.160$ |
| | C7 | C8 | $0.470 \pm 0.146$ | $0.483 \pm 0.127$ | $0.165 \pm 0.343$ | $0.629 \pm 0.436$ |
| | C1 | C3 | $0.467 \pm 0.089$ | $0.528 \pm 0.088$ | $2.1\text{e-}12 \pm 4.8\text{e-}11$ | $0.032 \pm 0.147$ |
| | C3 | C6 | $0.448 \pm 0.092$ | $0.474 \pm 0.087$ | $0.032 \pm 0.145$ | $0.076 \pm 0.224$ |

Table 6: The quantitative evaluation of boundary graphs generated by both our reproduction results of GNNBoundary and our baseline approach. The average predicted class probability of 1000 generated boundary graphs along with the corresponding standard deviation is presented below.

## E Three-way Adjacency Algorithm

---
**Algorithm 3** Measure the Degree of Adjacency of Three Classes

---
1: **count** $\leftarrow 0$
2: **for** $k \leftarrow 1 \ldots K$ **do**
3:     Randomly sample three graphs $G_{1_k} \in \mathcal{R}\{c_1\}$, $G_{2_k} \in \mathcal{R}\{c_2\}$, $G_{3_k} \in \mathcal{R}\{c_3\}$
4:     **for** $a, b \in [0, 1]$ where $a + b \leq 1$ **do**
5:         $c \leftarrow 1 - a - b$
6:         Compute interpolated embedding: $\phi_{interp} = a\phi_{L-1}(G_{1_k}) + b\phi_{L-1}(G_{2_k}) + c\phi_{L-1}(G_{3_k})$
7:         Store classification: $classes \leftarrow \arg\max_c \eta_{L-1}(\phi_{interp})$
8:     **end for**
9:     **if** $|classes| = 3$ **then**
10:         **count** $\leftarrow$ **count** $+ 1$
11:     **end if**
12: **end for**
13: **return** $\frac{\textbf{count}}{K}$

---

## F   Three-way adjacency and pairwise adjacency scores

| Dataset | Class Triplet | Three-way Score | Class Pair | Pairwise Score |
|---|---|---|---|---|
| Collab | HE-CM-Astro | 1.00 | HE-CM | 0.88 |
| | | | HE-Astro | 0.99 |
| | | | CM-Astro | 0.58 |
| Motif | H-HX-Cp4 | 1.00 | H-HX | 1.00 |
| | | | H-Cp4 | 1.00 |
| | | | HX-Cp4 | 0.04 |
| Motif | H-HX-Cp5 | 1.00 | H-HX | 1.00 |
| | | | H-Cp5 | 0.00 |
| | | | HX-Cp5 | 1.00 |
| Enzymes | EC1-EC5-EC6 | 0.72 | EC1-EC5 | 0.72 |
| | | | EC1-EC6 | 0.82 |
| | | | EC5-EC6 | 0.80 |
| Enzymes | EC1-EC2-EC6 | 0.61 | EC1-EC2 | 0.63 |
| | | | EC1-EC6 | 0.82 |
| | | | EC2-EC6 | 0.77 |
| Enzymes | EC2-EC5-EC6 | 0.69 | EC2-EC5 | 0.55 |
| | | | EC2-EC6 | 0.77 |
| | | | EC5-EC6 | 0.80 |
| Enzymes | EC2-EC3-EC6 | 0.63 | EC2-EC3 | 0.71 |
| | | | EC2-EC6 | 0.77 |
| | | | EC3-EC6 | 0.60 |
| MSRC9 | C6-C7-C8 | 1.00 | C6-C7 | 1.0 |
| | | | C6-C8 | 1.0 |
| | | | C7-C8 | 0.94 |
| MSRC9 | C1-C3-C5 | 0.98 | C1-C3 | 0.94 |
| | | | C1-C5 | 0.99 |
| | | | C3-C5 | 0.65 |
| MSRC9 | C2-C3-C4 | 0.97 | C2-C3 | 1.0 |
| | | | C2-C4 | 0.86 |
| | | | C3-C4 | 0.84 |
| MSRC9 | C2-C3-C7 | 0.94 | C2-C3 | 1.0 |
| | | | C2-C7 | 0.88 |
| | | | C3-C7 | 0.77 |
| MSRC9 | C3-C6-C8 | 0.93 | C3-C6 | 1.0 |
| | | | C3-C8 | 0.87 |
| | | | C6-C8 | 1.0 |
| MSRC9 | C2-C3-C8 | 0.92 | C2-C3 | 1.0 |
| | | | C2-C8 | 0.84 |
| | | | C3-C8 | 0.87 |
| MSRC9 | C4-C6-C8 | 0.92 | C4-C6 | 0.0 |
| | | | C4-C8 | 1.0 |
| | | | C6-C8 | 1.0 |

Table 7: Three-way and pairwise adjacency scores across different datasets.

