# OpenReview forum: "Reproducibility Study of GNNBoundary: Towards Explain- ing Graph Neural Networks through the Lens of Decision Boundaries"
_TMLR — Rejected by TMLR_

### Review · Reviewer_QqVo · 2025-03-12

**Summary Of Contributions:**

The present paper is a reproducibility study of the GNNBoundary paper by Wang & Shen (2024). GNNBoundary is a model-level explanation technique used to identify classes that are "adjacent" (similar) and generate instances that are close to the decision boundary of theses classes. These generated graphs can be used to analyze the boundary thickness, boundary margin and complexity of the margin. The paper reproduces the results of the original paper, adds the MSRC9 dataset to the evaluation.. The experimental results identify discrepancies with respect to stability and generalizability. Moreover, the authors identify specific hyperparameter choices that were not sufficiently discussed. Lastly, the authors propose an extension of the GNNBoundary approach to three-class-boundaries, and evaluate the corresponding class predictions in  this setting.

**Audience:**

Yes

**Broader Impact Concerns:**

I do not have any concerns on the ethical implications of this work.

**Claims And Evidence:**

Yes

**Requested Changes:**

As of now, I am leaning towards rejecting this paper due to the limited analysis. For acceptance, I recommend the following revisions:
- GNNBoundary also discusses the following contribution in their conclusion, which is not addressed in the reproducibility study.
> More importantly, our case studies illustrate that the boundary graphs generated by GNNBoundary can be utilized to measure the boundary thickness, boundary margin, and complexity of the boundary
- Three-way decision boundary graphs are not performing well. The paper should address this problem. Especially, some pairwise scores seem to be very low in Appendix F. As of now, the added value of the three-way decision boundary is quite limited. Moreover, I would assume that there is some sort of associative property between pairwise adjacent classes, which could be utilized to draw conclusions about the three-way adjacent classes. An isolated comparison with such a baseline could be beneficial.
- Adding additional datasets to the experimental analysis would strengthen the contribution of this paper.

**Minor**
- "reproduce" typo in Section 3, first sentence
- "Gumbell", page 5, bottom, 2. bullet point

**Strengths And Weaknesses:**

**Strengths**
- The paper is easy to follow and the objective is clearly stated
- Understanding the limitations of GNNBoundary is important
- The paper identifies several limitations of GNNBoundary with respect to stability, generalizability and specific hyperparameter choices. Theses are important findings
- The paper proposes as extensions to three-class-boundaries, which is an interesting direction.

**Weaknesses**
- The reproducibility study does not address the contribution of GNNBoundary to analyze the margin (see requested changes)
- Introduction, related work and background are **very** similar to the original paper, which add very little value.
- The three-class-boundary does not perform very well. Moreover, the three-way-boundaries are compared with high pairwise scores in Appendix F, which sometimes deteriorate.
- The paper examines one additional dataset, where it would be beneficial to expand the scope of this analysis to more datasets, e.g. from [1]

[1] Morris, Christopher, et al. "TUDataset: A collection of benchmark datasets for learning with graphs." ICML 2020 Workshop on Graph Representation Learning and Beyond (GRL+ 2020). 2020.

---

### Review · Reviewer_g8V4 · 2025-04-23

**Summary Of Contributions:**

**EDIT**: I changed the original review after realising that I had been falsely assuming that this work was a resubmission of a [previous work](https://openreview.net/forum?id=zLfLTHOdZW) I reviewed, due to their similarity.

The authors present a reproducibility study of "GNNBoundary: Towards Explaining Graph Neural Networks through the Lens of Decision Boundaries", which introduced a model-level explainability method for GNNs. They reproduce the original experiments, pointing out some reproducibility concerns, and introduce an extension to three-classes borders.

**Audience:**

Yes

**Claims And Evidence:**

Yes

**Requested Changes:**

I list here some of the minor issues, mainly typos:
- $R_c^{(l)}$ is first introduced as "decision regions" in the embedding space of each GNN layer, but later you also write $G\in R_c^{(l)}$, but the graph $G$ can't be in the embedding space.
- Similarly, $H^{(l)}$ is first defined as indicating the embedding space itself, while later it is used to indicate the embedding of a specific graph.
- As before, the expression $G_{c_1 || c_2}\in \mathcal{B}_{c_1||c_2}$ doesn't make sense, since $\mathcal{B}$ is a subset of the embedding space.
- Sometimes you miss the "\mathcal" command when referring to the decision boundary.
- From paragraph 5.1.1, the embeddings $H^{(l)}$ appear in bold.
- Expressions like "For a boundary graph $b\in\{c_1,c_2}$" are unclear. Is $b$ a graph or a class index?
- Sometimes, subscripts are not correctly formatted.
- In equation (2), $p^*$ is not defined.
- $\mathbb{\Psi}$ is not defined.
- This expression is unclear: $\Psi(G)=p(c_1),p(c_2)\in (p_{min},p_{max})(G)$.

The main problem of this work is when one compares it with the work cited above, which admittedly provides a deeper analysis of GNNBoundary.
The only element that sets them apart is the addition of the three-class case, although it doesn't seem to add much.

**Strengths And Weaknesses:**

Strengths:
- The paper is well structured and well written overall.
- Even though it is a reproducibility study, so you are expected to know the original paper, it is quite self-standing, providing a good explantion of GNNBoundary.
- The authors extend the study by applying the method to the three-class boundary case and to a new dataset.

Weaknesses:
- There are still some typos and unclear notations. I list below some examples.
- It's not clear, given that the authors themselves admitted that the three-class case "achieved moderate success", why this addition would be interesting. Also, isn't the three-class boundary just a special case of the two-class boundary? If not, why stop at 3 classes?
- The present work is very similar, in its scope and claims, to [this one](https://openreview.net/forum?id=zLfLTHOdZW). The cited paper adds further analysis:
    - They test the method on a new dataset but also on a new model (GAT).
    - They show that the boundary graphs generated by GNNBoundary only represent a small portion of the boundary.
    - They introduce and discuss the metrics provided in the original work (boundary thickness, margin and complexity). In my opinion, when reproducing a paper, one should also test the same metrics.

---

### Review · Reviewer_AxfA · 2025-04-29

**Summary Of Contributions:**

This study evaluates and extends the GNNBoundary method, which aims to explain Graph Neural Networks (GNNs) by analyzing decision boundaries between graph classes. I list several of the key contributions here:

Reproducibility Evaluation: The study assesses the reproducibility of key claims from the original GNNBoundary paper, including the identification of adjacent class pairs, the generation of boundary graphs, and the effectiveness of its adaptive loss function.

Extension to Three-Class Boundaries: The research extends the GNNBoundary method to handle three-class decision boundaries, exploring its feasibility and limitations in scenarios with more complex class interactions.

Performance Analysis: It provides a quantitative evaluation of the reproduced GNNBoundary and the new three-class extension across several datasets (Collab, Motif, Enzymes, MSRC9), comparing results to the original paper and a baseline.

Identification of Sensitivity: The study highlights the sensitivity of GNNBoundary's performance to hyperparameters and initialization, particularly noting challenges with convergence and variability in results.

**Audience:**

No

**Broader Impact Concerns:**

no.

**Claims And Evidence:**

No

**Requested Changes:**

See weakness part.

**Strengths And Weaknesses:**

Strengths:
Systematic Reproduction: The study systematically attempts to reproduce the key claims of the original GNNBoundary paper, following the described methodology for identifying adjacent classes and generating boundary graphs.
Extension Beyond Reproduction: The study goes beyond simple reproduction by proposing and implementing a methodological extension to handle three-class decision boundaries, including developing a novel three-way adjacency detection algorithm using barycentric interpolation.The evaluation uses the same datasets as the original paper (Collab, Motif, Enzymes) and introduces an additional, more complex dataset (MSRC9) to test scalability and robustness. The study reports computational requirements, training times (CPU vs. GPU), and estimates the carbon footprint of the experiments.

Weaknesses:
Challenges with Three-Class Convergence: The methodology for the three-class extension, including the adapted loss function and convergence criteria, did not consistently achieve balanced boundary graphs, suggesting the approach may require further methodological refinement.

You are mentioning that "Future work should explore methods to enhance classifier accuracy, particularly for datasets like Enzymes, where lower classification performance may lead to unreliable adjacency estimations. Analyzing the relationship between classifier confidence and adjacency scores could improve the robustness of GNNBoundary by mitigating sensitivity to noisy predictions". However, I think it is exactly what you need to mention/ at least give some insights in this paper.

---

### Decision · Action_Editor_82we · 2025-06-03

**Recommendation:** Reject

**Additional Comments:**

2 out of the 3 reviewers raised legitimate concerns in their reviews and recommend to not accept it in its current form. The authors did not answer to the reviews.

**Audience:**

No

**Audience Explanation:**

As mentioned by one of the reviewer "There is little added value in this reproducibility study, where the three-class case shows mixed results. Expanding the scope of this work and addressing the current limitations would be necessary" to make the work more relevant to TMLR's audience.

**Claims And Evidence:**

No

**Claims Explanation:**

As mentioned in the reviews, this reproducibility study doesn't discuss the metrics provided in the original work (boundary thickness, margin and complexity) and fails to justify the extension of the original work to the three-boundary case (which shows poor performances).

**Resubmission Of Major Revision:**

The authors may consider submitting a major revision at a later time.